# A Novel Multi-Task Learning Model with PSAE Network for Simultaneous Estimation of Surface Quality and Tool Wear in Milling of Nickel-Based Superalloy Haynes 230

**DOI:** 10.3390/s22134943

**Published:** 2022-06-30

**Authors:** Minghui Cheng, Li Jiao, Pei Yan, Huiqing Gu, Jie Sun, Tianyang Qiu, Xibin Wang

**Affiliations:** 1School of Mechanical Engineering, Beijing Institute of Technology, No. 5 Zhongguancun South Street, Haidian District, Beijing 100081, China; 18513389025@163.com (M.C.); chincock@163.com (H.G.); sj_bit@163.com (J.S.); 2Key Laboratory of Fundamental Science for Advanced Machining, Beijing Institute of Technology, No. 5 Zhongguancun South Street, Haidian District, Beijing 100081, China; jiaoli@bit.edu.cn (L.J.); tianyangqiu@bit.edu.cn (T.Q.); cutting0@bit.edu.cn (X.W.)

**Keywords:** multi-task learning, parallel-stacked auto-encoder, dynamic weight averaging, surface roughness estimation, tool wear estimation

## Abstract

For data-driven intelligent manufacturing, many important in-process parameters should be estimated simultaneously to control the machining precision of the parts. However, as two of the most important in-process parameters, there is a lack of multi-task learning (*MTL*) model for simultaneous estimation of surface roughness and tool wear. To address the problem, a new *MTL* model with shared layers and two task-specific layers was proposed. A novel parallel-stacked auto-encoder (PSAE) network based on stacked denoising auto-encoder (SDAE) and stacked contractive auto-encoder (SCAE) was designed as the shared layers to learn deep features from cutting force signals. To enhance the performance of the *MTL* model, the scaled exponential linear unit (SELU) was introduced as the activation function of SDAE. Moreover, a dynamic weight averaging (DWA) strategy was implemented to dynamically adjust the learning rate of different tasks. Then, the time-domain features were extracted from raw cutting signals and low-frequency reconstructed wavelet packet coefficients. Frequency-domain features were extracted from the power spectrum obtained by the Fourier transform. After that, all features were combined as the input vectors of the proposed *MTL* model. Finally, surface roughness and tool wear were simultaneously predicted by the trained *MTL* model. To verify the superiority and effectiveness of the proposed *MTL* model, nickel-based superalloy Haynes 230 was machined under different cutting parameter combinations and tool wear levels. Some other intelligent algorithms were also implemented to predict surface roughness and tool wear. The results showed that compared with the support vector regression (SVR), kernel extreme learning machine (KELM), *MTL* with SDAE (MTL_SDAE), *MTL* with SCAE (MTL_SCAE), and single-task learning with PSAE (STL_PSAE), the estimation accuracy of surface roughness was improved by 30.82%, 16.67%, 14.06%, 26.17%, and 16.67%, respectively. Meanwhile, the prediction accuracy of tool wear was improved by 46.74%, 39.57%, 41.51%, 38.68%, and 39.57%, respectively. For practical engineering application, the dimensional deviation and surface quality of the machined parts can be controlled through the established *MTL* model.

## 1. Introduction

Traditional manufacturing has been smoothly transformed into data-driven intelligent manufacturing because of the rapid development of intelligent sensing instruments, artificial intelligence, and industrial internet of things. Metal cutting is an important part of the manufacturing industry. There is an increasing demand to predict multiple in-process parameters to control the machined surface quality. Surface roughness and tool wear are two of the most important in-process parameters, which are related to the machining accuracy and efficiency of the parts. As an important evaluation index of machined surface quality, surface roughness affects the working state of mechanical parts in the assembly [1], the fatigue strength, and creep life of the parts [2,3]. In addition, the machining precision, machining efficiency, and surface integrity will be affected by the tool wear conditions. Severe tool wear may directly lead to the scrapping of production parts and machine downtime. Previous research has shown that the abnormal tool wear conditions will cause 20% of downtime [4]. Therefore, it is necessary to achieve both surface roughness and tool wear estimation toward precision machining.

Many models or methods have been proposed for surface roughness and tool wear estimation. Estimation models related to surface roughness were approximately divided into three categories: models based on the machining theory, models established with multiple linear regression empirical formula, and data-driven prediction models [5]. Similarly, machining-theory-based models and data-driven models have been established for tool wear estimation [6]. Considering the validity and applicability, data-driven prediction models are the most suitable for current data-driven manufacturing environments, as they do not rely on complex cutting theory and can be integrated with the manufacturing systems.

For data-driven prediction models, the mainstream research process mainly includes the following steps: cutting signals collection, feature extraction and selection, data-driven model establishment and training, and model validation. For example, Huang et al. realized the surface roughness prediction with gray system theory and the time-domain features extracted from cutting force signals [7]. Du et al. achieved the estimation of multiple surface quality characterization parameters with a machine-learning model and multiple cutting signals [8]. Li et al. constructed a tool wear prediction model with meta-learning and physics-informed model [9]. Qiao et al. proposed a parallel deep-learning structure for feature extraction and realized tool wear estimation with multiple sensors fusion [10]. These studies fully demonstrated the effectiveness and prospects of artificial intelligence models combined with intelligent sensing devices in tool wear and surface roughness estimation. Furthermore, some researchers considered tool wear an unavoidable factor during the machining process. Thus, tool wear was also considered as the input of the surface roughness estimation models. Li et al. established a surface roughness prediction model with the least squares support vector machine (LSSVM), and the basic cutting parameters and tool wear were combined as the input variables [11]. Agustina et al. evaluated the effect of tool wear on the surface roughness [12]. Pimenov et al. adopted the random forest model to predict the surface roughness under the consideration of tool wear [13]. In addition, some other researchers believed that tool wear states would be reflected on the surface topography, so the surface roughness was used as the input data to establish the tool wear estimation models. For example, energy, entropy, inertial moment, and correlation were extracted from surface topography, and then, the tool wear states were estimated through these features [14,15]. Jain et al. established a tool wear monitoring model with the current and previous surface roughness values as the input variables [16]. Kene et al. established a tool health monitoring system using cutting force and surface roughness signals [17]. The above studies fully demonstrated the strong correlation between surface roughness and tool wear. In addition, as important in-process parameters, tool wear is directly related to dimensional deviation of the machined parts [18], and surface roughness is important in evaluating the machined surface quality. Therefore, both surface roughness and tool wear should be estimated simultaneously. However, these studies did not establish unified models for simultaneously predicting surface roughness and tool wear. Wang et al. proposed a hybrid deep-learning model for tool condition prognosis, and surface roughness was also predicted based on the monitored tool wear [19]. Rao et al. established an optimized gray model (OGM) (1,N) to predict surface roughness and tool wear simultaneously using vibration signals [20]. Huang et al. estimated tool wear and surface roughness simultaneously with one-dimensional convolutional neural network (1D CNN) and multiple sensors fusion method [21]. Similarly, Chen et al. studied the applications of 1D CNN and vibration signals in tool wear detection and surface roughness estimation [22]. Although those data-driven models have achieved simultaneous prediction of surface roughness and tool wear, the established models were still single-task prediction models. In other words, the common knowledge and complementary information of surface roughness and tool wear were not considered.

To fully utilize the correlation and common information between surface roughness and tool wear, a joint prediction model should be employed to explore the common information [23]. As a type of joint prediction model, the multi-task learning (*MTL*) model has been developed to improve the generalization ability of a single task through shared representation and implicit data augmentation [24]. Due to the information-sharing mechanism focusing on the mutual effective features and simultaneous estimation of multiple related tasks, *MTL* models have been successfully applied to gas detection and concentration estimation [25], Alzheimer’s disease progression detection [26], and maintenance of mechanical system health status [27]. For the practical machining process, Liu et al. established a *MTL* model with the deep neural network for accurate prediction of tool wear and chipping [28]. Wang et al. constructed a *MTL* model with the deep belief network (DBN) for identifying surface quality and tool wear condition [29]. Although both surface roughness and tool wear were considered, only the classification problem was solved. Moreover, the training process of DBN is time consuming with the contrastive divergence algorithm, and the stochastic probability mechanism that existed in DBN is prone to unstable results [30]. Therefore, a suitable *MTL* model with robust performance needs to be proposed to solve the regression problem of surface roughness and tool wear estimation.

Based on the understanding of the above studies, a number of data-driven models with intelligent sensors have been proposed for tool wear monitoring and surface roughness estimation, and satisfactory results were obtained. These studies have made important contributions to the development of intelligent, networked, and digitalized machining processes, in line with the requirements of Industry 4.0 [31]. However, these studies mainly focused on the development and application of intelligent sensing devices and advanced machining algorithms [32]. Surface roughness prediction and tool wear monitoring were regarded as two independent tasks. The complementary information and common knowledge of multiple in-process parameters in the machining process were ignored. Moreover, the machine-learning algorithms used can be divided into two categories: shallow machine-learning models, such as artificial neural network, support vector machine, and gray model; and deep-learning models, such as CNN and recurrent neural networks (RNN). Nevertheless, shallow machine-learning models cannot learn deep features from input vectors, and the estimation accuracy may not be satisfactory. The deep-learning models similar to CNN can learn deep features, but they have requirements on the amount of experimental data. Collecting sufficient data is a time-consuming and costly process. Therefore, unsupervised or semi-supervised algorithms should be fully used to fill in this limitation [33]. To solve these problems, a multi-task learning model was proposed to fully mine and exploit complementary information and common knowledge of multiple in-process parameters. A novel unsupervised algorithm named parallel-stacked auto-encoder network was developed to learn deep features related to surface roughness and tool wear, and it was used as the shared layers of the *MTL* model. Eventually, not only surface roughness and tool wear estimation were achieved simultaneously, but also the accuracy of both was significantly improved.

Overall, a novel *MTL* model with shared layers and two task-specific layers was proposed for the simultaneous estimation of surface roughness and tool wear. Firstly, the effective features were extracted from the original cutting force signals, power spectrum, and reconstructed low-frequency wavelet packet coefficients, which were fed into the proposed *MTL* model. To learn deep features from the extracted statistical features, a new parallel-stacked auto-encoder (PSAE) based on a three-layer stacked denoising auto-encoder (SDAE) and a three-layer stacked contractive auto-encoder (SCAE) was developed as the shared layers. Two dense layers were employed as the task-specific layers for surface roughness and tool wear estimation. Furthermore, a scaled exponential linear unit (SELU) was introduced as the activation function of SDAE to further improve the efficiency of the *MTL* model. Finally, to balance the contribution degree of different tasks, the learning rate of different tasks was dynamically adjusted through the dynamic weight averaging (DWA) strategy.

## 2. The Proposed Multi-Task Learning Model

Multi-task learning (*MTL*) model was proposed to enhance single-task generalization ability by sharing the common knowledge in the related tasks. The essence of *MTL* lies in parameter sharing. Currently, the parameter sharing strategies mainly include hard parameter and soft parameter sharing. Hard parameter sharing is suitable for multiple related tasks. By comparison, soft parameter sharing is more flexible and does not require task relevance, but many additional parameters will be added, which will bring a burden to the model training. Considering the strong correlation between surface roughness and tool wear [18], a novel *MTL* model based on the hard parameter sharing principle was proposed in this paper.

### 2.1. Multi-Task Network Structure with Parallel-Stacked Auto-Encoder

The proposed *MTL* model with hard parameter sharing mainly includes shared layers and task-specific layers. The shared layers were designed to share and complement the domain-specific information, and the task-specific layers were designed to learn specific features for different tasks. In this paper, a novel unsupervised network structure called PSAE based on SDAE and SCAE was developed as the shared layers. Two dense layers were employed as the task-specific layers. Then, the time-domain features were calculated from raw cutting signals and reconstructed wavelet packet coefficients with a low frequency. The power spectrum obtained by the Fourier transform (FT) was used to extract frequency-domain features. Finally, all features were merged as the input data of the proposed *MTL* model.

#### 2.1.1. Standard Stacked Auto-Encoder

A standard stacked auto-encoder is constructed by stacking several auto-encoders. The auto-encoder (AE) belongs to the unsupervised learning algorithm, mainly including the encoder and decoder, as shown in Figure 1. For a given training sample $x=[x_{1},x_{2},x_{3},\ldots ,x_{n-1},x_{n}]$, the feature representation $h=[h_{1},h_{2},\ldots ,h_{m-1},h_{m}]$ is obtained through nonlinear mapping in the encoding step. The output vector $y=[y_{1},y_{2},y_{3},\ldots ,y_{n-1},y_{n}]$ is reconstructed to fit the input data x by minimizing the loss function in the decoding step. The specific encoding and decoding operations are expressed as
(1)$$h=f(W_{E}x+b_{E})$$
(2)$$y=f(W_{D}h+b_{D})$$
where $\left\{W_{E},\;b_{E}\right\}$ and $\left\{W_{D},\;b_{D}\right\}$ denote the parameter set of the encoder and decoder, respectively, and *f* represents the activation function, which determines the mode of feature transformation of the AE. However, there is no mature theory for determining a suitable activation function. According to previous research [34,35], Sigmoid, Tanh, SoftPlus, exponential linear unit (ELU), and rectified linear unit (RELU) are commonly used to design deep neural networks. In addition to these five activation functions, the scaled exponential linear unit (SELU) is also introduced as the activation function in this paper [36]. The feature representation of the input data mapped by SELU can well meet normalization distribution, which accelerates the convergence speed. The waveforms of the six activation functions are shown in Figure 2.

Then, the parameter set $\theta =\{W_{E},b_{E},W_{D},b_{D}\}$ of *AE* is adjusted to minimize the reconstruction error between ***x*** and ***y***. In this paper, the reconstruction error was calculated in terms of the mean squared error, which is expressed as
(3)$$L_{AE}=\frac{1}{n}{\sum_{i=1}^{n}\left\|x_{i}-y_{i}\right\|}^{2}$$
where *n* represents the number of the training datasets, and $L_{AE}$ denotes the loss function (reconstruction error). To learn deep feature representation from the extracted features, the stacked auto-encoder (SAE) is constructed with several trained auto-encoders, as shown in Figure 3. To be specific, the feature representation ***h***_1_ of AE-1 is sent to the input layer of the contiguous AE-2. Similarly, the feature representation ***h***_2_ of AE-2 is fed into the input layer of the AE-3. Then, the SAE is obtained by a layer-wise construction process.

#### 2.1.2. SDAE and SCAE

In practical applications, the training dataset samples are often mixed with some noise, leading to the loss of data authenticity. Therefore, the feature representation obtained from SAE contains some errors due to the noise. To solve this problem, a denoising auto-encoder (DAE) was proposed by adding a certain amount of noise to the input data, and then, the robust feature representation of input data is learned. The corresponding formula is shown as follows:(4)$$\hat{x}=x+noise(x)$$

Moreover, to enhance the ability to learn feature representation, a contractive auto-encoder (*CAE*) was developed by adding a contractive penalty term to the loss function [31]. Additionally, the modified reconstruction error is expressed as
(5)$$L_{CAE}=\frac{1}{n}{\sum_{i=1}^{n}\left\|x_{i}-y_{i}\right\|}^{2}+\lambda {\left\|J_{w}(x)\right\|}^{2}$$
where λ is the regularization coefficient parameter. $J_{w}(x)$ represents the Jacobian matrix of the encoder, which is expressed as
(6)$${\left\|J_{w}(x)\right\|}^{2}={\left\|\frac{\partial w}{\partial x}\right\|}^{2}={\sum_{j=1}^{m}\sum_{i=1}^{n}\left(\frac{\partial w}{\partial x}\right)}^{2}$$
(7)$$J_{w}(x)=\left[\begin{array}{llll} \frac{\partial w_{1}}{\partial x_{1}} & \frac{\partial w_{1}}{\partial x_{2}} & \cdots & \frac{\partial w_{1}}{\partial x_{n}}\\ \frac{\partial w_{2}}{\partial x_{1}} & \frac{\partial w_{2}}{\partial x_{2}} & \cdots & \frac{\partial w_{2}}{\partial x_{n}}\\ \vdots & \vdots & \vdots & \vdots \\ \frac{\partial w_{m}}{\partial x_{1}} & \frac{\partial w_{m}}{\partial x_{2}} & \cdots & \frac{\partial w_{m}}{\partial x_{n}} \end{array}\right]$$

Similar to the construction process of SAE, SDAE and SCAE are constructed through the layer-wise construction process.

#### 2.1.3. The Overall Framework of the MTL Model

Generally, the individual stacked auto-encoder is prone to poor performance and low generalization in dealing with diverse and complex cutting signals. To overcome the limitations of the individual stacked auto-encoder, ensemble learning is regarded as an effective technique to improve generalization and robustness by combining different stacked auto-encoders [35,37]. Ensemble learning is not a machine-learning algorithm, but it completes the learning task by combining several learners. Therefore, combining the complementary advantages and characteristics of SDAE and SCAE, the PSAE based on the three-layer SDAE and three-layer SCAE is proposed to learn more effective features, and it is used as the shared layers. To learn specific features for individual tasks, a set of dense layers are designed as task-specific layers. Moreover, when the machining process runs under variable working conditions, the amplitude of cutting signals and the degradation pattern of the cutting tool are affected by the variable working conditions. Therefore, the basic cutting parameter information is also considered to improve the accuracy of surface roughness and tool wear prediction. To obtain effective features, the cutting parameter information is further processed by a dense layer, which is also used as the shared layers. After that, the deep features learned by the PSAE are concatenated with the effective features obtained from the basic cutting parameters to form hybrid information. Then, the hybrid information is fed into two parallel task-specific layers for simultaneous prediction of surface roughness and tool wear, as shown in Figure 4.

The parameters in the *MTL* model that need to be optimized include two parts: shared and task-specific parameters. To achieve joint optimization of all parameters, the optimization goal is defined as the weighted sum of loss functions of different tasks as follows
(8)$$L_{MTL}=\lambda_{1}\frac{1}{N_{b}}\sum_{i=1}^{N_{b}}{\left(S_{a_{i}}{}^{pre}-S_{a_{i}}\right)}^{2}+\lambda_{2}\frac{1}{N_{b}}\sum_{i=1}^{N_{b}}{\left(VB_{\max}{{}_{i}}^{pre}-VB_{\max}{}_{i}\right)}^{2}$$
where $L_{MTL}$ represents the loss function of the *MTL* model, $\lambda_{1}$ and $\lambda_{2}$ denote the loss weight of surface roughness and tool wear, respectively; $N_{b}$ represents the number of samples in each batch during model training, $S_{a}{}^{pre}$ and $VB_{\max}{}^{pre}$ represent the surface roughness and tool wear values obtained by the *MTL* model, respectively; $S_{a}$ denotes the actual surface roughness value, and *VB*_max_ represents the measured tool wear. Due to high computational efficiency and low memory footprint, the Adam algorithm is selected to adjust all parameters in the *MTL* model. In addition, the hyper-parameters of the *MTL* model, such as the number of hidden layer nodes in the AE and the number of epochs, need to be determined to improve the accuracy of the model. Similarly, with Equation (8) as the optimization objective, the hyper-parameters were determined using the validation dataset with the help of the grid search framework. Grid search can obtain satisfactory results in low-dimensional space and is easily applied to deep-learning model framework [38]. Its main idea is that if there are k hyper-parameters that need to be optimized, the values are taken at equal intervals according to the range of the hyper-parameters. Then, all parameter combinations are considered in a searching loop manner (grid format). The hyper-parameter combination of the model was determined when the loss of the validation dataset was minimal, as shown in Table 1.

### 2.2. Dynamic Weight Averaging

For the *MTL* model, the weights $\lambda_{1}$ and $\lambda_{2}$ of the multi-task loss function are used to measure the relative contribution of each task. All tasks should be made equally important, rather than allowing learning to be dominated by easier tasks. Without finding a suitable balance between these tasks, it is difficult to learn multiple tasks. Therefore, dynamic weight averaging (DWA) is introduced to dynamically adjust the learning rate of different tasks [39]. According to the loss function value of surface roughness and tool wear during the iterative process, the mathematical model of DWA is expressed as
(9)$$w_{k}(t:t+4)=\frac{\sum_{i=t-5}^{t-1}L_{k}(i)}{\sum_{j=t-10}^{t-6}L_{k}(j)}$$
(10)$$\lambda_{k}(t:t+4)=\frac{K\cdot \exp(w_{k}(t:t+4))}{\sum_{k=1}^{2}\exp(w_{k}(t:t+4))}$$
where *K* represents the number of tasks to be jointly learned, *t* represents the current iteration number, and $L_{k}(.)$ denotes the loss function of the *k*-th task. To reduce the uncertainty from stochastic gradient descent, the loss value $L_{k}(.)$ is calculated as the average loss over five consecutive loss function values. $w_{k}(.)$ represents the relative descending rate of the *k*-th task. $\lambda_{k}(.)$ denotes the adjusted weight of the *k*-th task, and the initial weights for surface roughness and tool wear are set to 1. Then, the learning rate of multiple tasks is adjusted and balanced during model training with the help of DWA.

## 3. Experimental Setup

To illustrate the efficiency of the proposed *MTL* model, milling experiments were carried out on the five-axis machining center DMU 80 mono-BLOCK (DMG, Munich, Germany). Due to excellent high temperature strength, corrosion resistance, and long-term thermal stability, Haynes 230 is employed as the main material of gas turbine combustor shells, which have a high requirement for surface quality. In addition, the cutting tool is prone to wear during the machining process because of the low thermal conductivity and adhesion phenomenon between the workpiece material and cutting tool. Thus, Haynes 230 was determined as the workpiece to be machined, and its mechanical properties and chemical compositions are listed in Table 2 and Table 3. Among them, the hardness of the material was measured by the Vickers hardness tester MHVS-30AT with a load of 0.5 kg (4.9 N) for 10 s.

The cutting tool used for machining Haynes 230 is XOMX090308TR-M08 F40M coated with PVD, and the cemented carbide is served as the matrix material. Its geometry is shown in Table 4. To study the correlation between tool wear and surface roughness, two cutting tools with a maximum flank wear of 100 μm were prepared in advance. A total of four cutting tools were used in the experiment, including two fresh cutting tools and two cutting tools with a maximum flank wear of 100 μm. In addition, considering the structural constraints of the parts and the difference in machining requirements, cutting parameters may need to be dynamically changed in the actual production process. Thus, the milling experiments were carried out under different cutting parameter combinations and tool wear levels. To facilitate machining, and considering that tool wear is not sensitive to the change of cutting width, the value of cutting width was fixed at 15 mm. In addition, according to the recommended parameters of the production department of the gas turbine combustor shells and preliminary experiments, the variation ranges of the other cutting parameters were determined with the cutting tool life as the evaluation standard, as shown in Table 5.

During the machining process, the workpiece Haynes 230 with a size of 70 mm × 60 mm × 25 mm was mounted on a three-component dynamometer (Kistler 9257B, Switzerland), and the cutting force signals were collected with a sampling frequency of 5 kHz (Fx, Fy, and Fz), as shown in Figure 5a. End milling was adopted with each cutting stroke of 70 mm. The cutting width was 15 mm, so a complete surface was milled four times (60/15 = 4), as shown in Figure 5b.

After machining, the maximum wear width near the tool tip and the machined surface topography were measured with the confocal laser scanning microscope (KEYENCE VK-100, Japan). According to the ISO 3685, the typical pattern of tool flank wear is shown in Figure 6a. The wear is relatively uniform in the middle part of the cutting edge involved in cutting (area B), and VB is often utilized to represent the average flank wear width. The maximum flank wear width is represented by *VB*_max_. However, in our experiment, the wear area of the flank wear is distributed in a circular arc, and there is no obvious uniform wear area. Moreover, the maximum tool flank wear is often used as the criteria to define the end of effective tool life [40]. Therefore, the maximum wear width near the tool tip was measured, as shown in Figure 6b. Considering that the machined surface is not an ideal regular surface, some adhered material particles, plowing grooves, and scratch marks may exist in the machined surface [41]. In addition, tool wear was considered in the experiment, and irregular vibration caused by the tool wear will also be reflected on the machined surface. If two-dimensional surface roughness *Ra* is used as the evaluation index, the surface roughness value at different positions of the machined surface may vary greatly. To avoid randomness and accidental errors, the three-dimensional surface roughness *Sa* was calculated from the scanned surface topography based on ISO 25178. The measurement positions of the surface topography are shown in Figure 5c. To avoid the effect of vibration upon tool entry and exit, the measurement positions are concentrated in the middle of the machined surface, and each measurement position is 10 mm apart. After determining the measurement position, the target surface was scanned with the microscope equipped with a laser to record the spatial distribution. In addition, to improve the measurement accuracy, the surface topography at different positions under the same set of cutting parameters was measured three times, and the averaged surface roughness value was taken as the final value.

To judge the stability of the milling process and the rationality of the cutting parameters, a modal test was carried out on our machining system. An MSC-1 impact hammer with a 500 kgf sensor was used to generate stimulus signals. A YD 67 acceleration sensor with a sensitivity of 0.38 PC/(m.s^−2^) was adhered to the selected position of the cutting tool to obtain response signals, as shown in Figure 7. According to the frequency response function in the X and Y direction, the approximate natural frequency of the machine tool–tool system was 2000 Hz.

Natural frequency plays an important role in judging whether chatter occurs during machining. The dominant chatter frequency is always near the natural frequency of the tool system and can be easily detected by the frequency spectrum. Therefore, the fast Fourier transform (FFT) was performed on the cutting force signals. In general, the greater the cutting depth, the more likely that chatter will occur. Thus, taking the cutting force signals with a cutting speed of 90 m/min, a feed per tooth of 0.05 mm/tooth, a cutting depth of 0.4 mm as an example, the frequency domain analysis was carried out, as shown in Figure 8. The diameter of the tool holder used in our experiment was 20 mm, so the spindle speed was n = 1432 r/min, and the spindle rotation frequency (domain frequency) was f = n/60 = 23.87 Hz.

From the frequency spectrum of cutting forces in three directions, the frequencies mainly focus on domain frequency and its multiple frequency. No frequencies close to natural frequency were observed. The results show that the milling process is stable, and the selection of the cutting parameters is reasonable.

Due to the high sampling frequency of the cutting force, if the cutting force signals are directly fed into the *MTL* model, the modeling process of the *MTL* model is time consuming. Thus, effective features are calculated from the original cutting signals to preserve the main characteristics of the cutting signals and reduce the input dimension of the *MTL* model. According to previous studies [42,43,44], a total of fourteen features were extracted, as shown in Table 6.

Because of a large difference in the magnitudes of the calculated features, normalization technique is needed to improve the performance of the *MTL* model. The Max–Min normalization method is adopted to normalize the extracted features, and its expression is shown as follows:(11)$$Fea_{norm}^{i}=\frac{Fea^{i}-\min(Fea^{i})}{\max(Fea^{i})-\min(Fea^{i})}$$
where $Fea^{i}$ represents the *i*th extracted feature.

To fully exploit the features hidden in the cutting force signals, wavelet packet transform (WPT) was also performed on the cutting signals, and the schematic diagram of the four-level WPT was shown in Figure 9. In the first level, A_1_ represents the approximation coefficients, and D_1_ is the detail coefficients. Starting from the second level, if the initial letter is A, it represents the approximation coefficients obtained by the decomposition of the previous level. Similarly, if D is the initial letter, it represents the detail coefficients.

Traditional wavelet decomposition only decomposes the low-frequency parts of the cutting signals, while WPT decomposes both low- and high-frequency signals, which is a more delicate feature extraction method. Previous research showed that the features extracted from four-level WPT are more effective for surface roughness estimation [45]. Therefore, four-level WPT was performed on the original cutting force signals, and then, decomposed wavelet packet coefficients were reconstructed. Taking the Fz component force in the Z direction as an example, at a cutting speed of 70 mm/min, a feed per tooth of 0.08 mm/tooth, a cutting depth of 0.2 mm, the reconstructed wavelet packet coefficients of the first four nodes are shown in Figure 10. The amplitude and overall trend of the reconstructed wavelet packet coefficients in the first node are basically consistent with the original cutting force signals. Starting from the second node, the waveforms of the wavelet packet coefficients are closer to the noise signals. In addition, the power spectrum of the first node contains only two main low-frequency components, whereas from the second node, the power spectrum starts to become chaotic. This objectively shows that the wavelet packet coefficients of the first node are related to the actual cutting process, and the wavelet packet coefficients of the other nodes are mainly determined by the cutting noise signals. Considering that the frequency of tooth passing and spindle rotation belongs to low frequency, the frequency range of AAAA_4_ is 0~156.25 Hz according to previous research [45], which contains the frequency of tooth passing and spindle rotation. Therefore, time-domain features were extracted only from the reconstructed wavelet packet coefficients AAAA_4_.

## 4. Results and Discussion

### 4.1. Correlation Analysis between Surface Roughness and Tool Wear

To determine the degree of correlation between surface roughness and tool wear, the Pearson correlation coefficient (*PCC*) was calculated. In statistics, *PCC* is employed to analytically evaluate the linear correlation between two continuous variables, and the expression is given as follows:(12)$$PCC=\frac{\sum \left(S_{a_{i}}-\bar{S_{a}})(VB_{\max}{}_{i}-\bar{VB_{\max}}\right)}{\sqrt{{\left(S_{a_{i}}-\bar{S_{a}}\right)}^{2}{\left(VB_{\max}{}_{i}-\bar{VB_{\max}}\right)}^{2}}}$$
where $\bar{S_{a}}$ is the mean value of surface roughness, and ${\bar{VB}}_{\max}$ is the mean value of tool wear. The value of *PCC* ranges from −1 to 1. The larger the absolute value of *PCC*, the stronger the correlation between two continuous variables. The variation trend of surface roughness versus tool wear is shown in Figure 11, and the value of *PCC* is 0.7022.

The value of *PCC* indicates that the surface roughness and tool wear have a strong correlation according to the judgment criteria. This theoretically illustrates that the proposed *MTL* model based on hard parameter sharing is suitable for the simultaneous estimation of surface roughness and tool wear.

### 4.2. The Selection of Activation Function for AE

The selection of the activation function has a significant influence on the performance of *AE*. To improve the performance of the proposed *MTL* model, a new type of activation function called SELU was introduced. Two different performance indicators were introduced to quantitatively evaluate the efficiency of the proposed *MTL* model with DWA, mainly including the mean absolute error (*MAE*) and root mean squared error (*RMSE*). The expressions are shown as follows:(13)$$\begin{array}{l} MAE=\frac{1}{N}\sum_{i=1}^{N}\left|Va_{\exp,i}-Va_{pre,i}\right|\\ RMSE=\sqrt{\frac{1}{N}\sum_{i=1}^{N}{\left(Va_{\exp,i}-Va_{pre,i}\right)}^{2}} \end{array}$$
where $Va_{\exp,i}$ denotes the actual surface roughness value or tool wear value, and $Va_{pre,i}$ represents the estimated surface roughness value or tool wear value by the trained *MTL* model. *N* is the sample size of the testing dataset. The detailed results with different activation functions are shown in Figure 12 and Figure 13, respectively.

As can be seen from Figure 12 and Figure 13, for different variants of *AE*, the optimal activation function is different. For the simultaneous prediction of multiple in-process parameters in the machining field, the best activation function for the denoising auto-encoder is SELU, while the optimal activation function for the contractive auto-encoder is RELU. In addition, compared with the most commonly used activation functions Sigmoid and RELU, when SDAE adopted the nonlinear function SELU as the activation function, the estimation accuracy of surface roughness was improved by 31.25% and 19.71%, respectively. Additionally, the estimation accuracy of tool wear was improved by 52.87% and 47.44%, respectively, as shown in Figure 12. This can be explained, as SELU has self-normalization and strong regularization ability [46]. The adverse effects caused by abnormal samples are mitigated through the normalization ability of SELU. Furthermore, the negative semi-axis of SELU is no longer set to 0, which solves the problem of nerve death in RELU [47]. It is worth noting that when SCAE adopts SELU as the activation function, the performance of the model has a downward trend. This reflected that the performance of SELU will be influenced by the loss function of the established neural network.

### 4.3. The Selection of Mother Wavelet for WPT

The choice of mother wavelet is an important issue in WPT. Some criteria have been proposed for selecting mother wavelet [42,48], such as maximum energy, minimum entropy, permutation entropy, but each selection criterion results in different mother wavelets. To determine the appropriate mother wavelet for WPT, five commonly used mother wavelets were selected from each wavelet family (Daubechies, Symlet, Coiflet, and Bior) according to previous studies [45,48,49], as shown in Table 7. A total of 20 mother wavelets were selected to study their impact on the multi-task learning model. Then, the optimal mother wavelet was determined by trial and error, and the performance of the *MTL* model under different mother wavelets was shown in Figure 14.

As can be seen from Figure 14, for our engineering problem, mother wavelet db3 is most suited for the *MTL* model, followed by db10. When mother wavelet coif4 is implemented for WPT, the prediction accuracy of the surface roughness is the highest, but the tool wear prediction accuracy is not ideal. Finally, mother wavelet db3 is used in this study.

### 4.4. Performance Evaluation of the Proposed MTL Model

To verify the superiority and effectiveness of the proposed *MTL* model, several other models are also used to predict surface roughness and tool wear, mainly including the support vector regression (SVR), kernel extreme learning machine (KELM), multi-task learning model only with the stacked denoising auto-encoder as the shared layers (MTL_SDAE), multi-task learning model only with the stacked contractive auto-encoder as the shared layers (MTL_SCAE), and single-task learning model with the proposed parallel-stacked auto-encoder (STL_PSAE).

**SVR:** As a conventional machine-learning model, it is often utilized to solve regression problems. The performance of SVR is sensitive to the selection of the kernel function coefficient and the penalty coefficient. Therefore, an enhanced whale optimization algorithm (EWOA) with an adaptive nonlinear position updating mechanism and differential mutation operation is employed to optimize these two coefficients. The whale optimization algorithm belongs to the swarm intelligence optimization algorithm, and its specific implementation can be referred to in the literature [50].

**KELM:** KELM is a single hidden layer feed-forward neural network, firstly proposed by Huang et al. [36]. Compared with the traditional neural network, KELM does not require backpropagation to adjust the parameters, and it has a better convergence speed and generalization performance. Similar to SVR, the performance of KELM is sensitive to the selection of the regularization parameter and the kernel parameter. Therefore, the EWOA is also utilized to determine the optimal values of these two parameters.

**MTL_SDAE** and **MTL_SCAE:** To verify the superiority of the proposed PSAE structure, only the SDAE or SCAE are used as the shared layers. Other parts of the structure and parameters of the models are the same as the proposed *MTL* model.

**STL_PSAE:** To verify the effectiveness of the multi-task learning structure, only one task-specific layer is designed for surface roughness or tool wear prediction. Other parts of the structure and parameters of the models are the same as the proposed *MTL* model.

In the current milling experiments, 114 sets of data were obtained. To verify the efficiency of the proposed *MTL* model, 20% of the dataset was randomly selected, served as the testing dataset, and the remaining data were utilized as the training dataset for training the models. After the models’ training was finished, the testing dataset was implemented to verify the performance of those trained models, and the surface roughness and tool wear estimation results were shown in Figure 15 and Figure 16, respectively.

As can be seen from Figure 15 and Figure 16, both in surface roughness and tool wear, the performance of the proposed *MTL* model outperforms SVR, KELM, MTL_SDAE, MTL_SCAE, and STL_PSAE. The estimation results of the proposed *MTL* model are very close to the actual values, especially for the tool wear estimation results. According to the characteristics of the multi-task learning framework, the results can be explained by the following three reasons. Firstly, the common knowledge and complementary information of surface roughness and tool wear are fully explored through the shared layers. Secondly, the multi-task learning structure can be regarded as a penalty term, which makes the model less prone to overfitting, leading to more stable and general prediction results [51]. Thirdly, the learning rate of surface roughness and tool wear is adjusted dynamically with the DWA strategy. In addition, the values of *MAE* and *RMSE* were calculated, as shown in Figure 17 and Table 8.

Among those models, the *MAE* and *RMSE* of the proposed *MTL* model are minimal, which illustrates the efficiency of the proposed *MTL* model for simultaneously improving the prediction accuracy of multiple related tasks. As shown in Table 8, in terms of *MAE*, compared with SVR, KELM, MTL_SDAE, MTL_SCAE, and STL_PSAE, the estimation accuracy of surface roughness is improved by 30.82%, 16.67%, 14.06%, 26.17%, and 16.67%, respectively. Meanwhile, the estimation accuracy of tool wear is improved by 46.74%, 39.57%, 41.51%, 38.68%, and 39.57%, respectively. For the actual production process, when the surface roughness of the machined surface exceeds production quality requirements, it is also necessary to replace the cutting tool in time [52]. Therefore, the proposed *MTL* model is more suitable for practical applications to monitor multiple in-process parameters simultaneously.

## 5. Conclusions

For data-driven intelligent manufacturing, it is significant to monitor multiple in-process parameters simultaneously. In this paper, a novel *MTL* model with shared layers and two task-specific layers was proposed for the simultaneous prediction of surface roughness and tool wear. The complementary information and common knowledge of surface roughness and tool wear were fully mined and exploited through the established *MTL* model. To learn deep features, a new unsupervised algorithm named the PSAE network structure based on SDAE and SCAE was proposed as the shared layers. Moreover, the DWA strategy was introduced to balance the learning rate of different tasks. Compared with previous studies, surface roughness and tool wear were simultaneously predicted, and the accuracy of both was significantly improved. Moreover, as an unsupervised algorithm, PSAE achieved satisfactory results even under small-sample data.

The superiority and effectiveness of the proposed *MTL* model were verified with milling experiments under different cutting parameter combinations and tool wear levels. Taking *MAE* as the quantification indicator, compared with SVR, KELM, MTL_SDAE, MTL_SCAE, and STL_PSAE, the prediction accuracy of surface roughness was improved by 30.82%, 16.67%, 14.06%, 26.17%, and 16.67%, respectively. Meanwhile, the prediction accuracy of tool wear was improved by 46.74%, 39.57%, 41.51%, 38.68%, and 39.57%, respectively. The proposed *MTL* model is effective for solving the regression prediction problems of multiple in-process parameters. In the future, multi-task learning can be combined with transfer learning to solve the simultaneous estimation of multiple in-process parameters under variable working conditions with limited data. Moreover, more effective features related to tool wear and surface roughness can be extracted from multiple sensors. The fusion of multi-source sensors is worth studying to improve prediction accuracy.

## Figures and Tables

**Figure 1 sensors-22-04943-f001:**
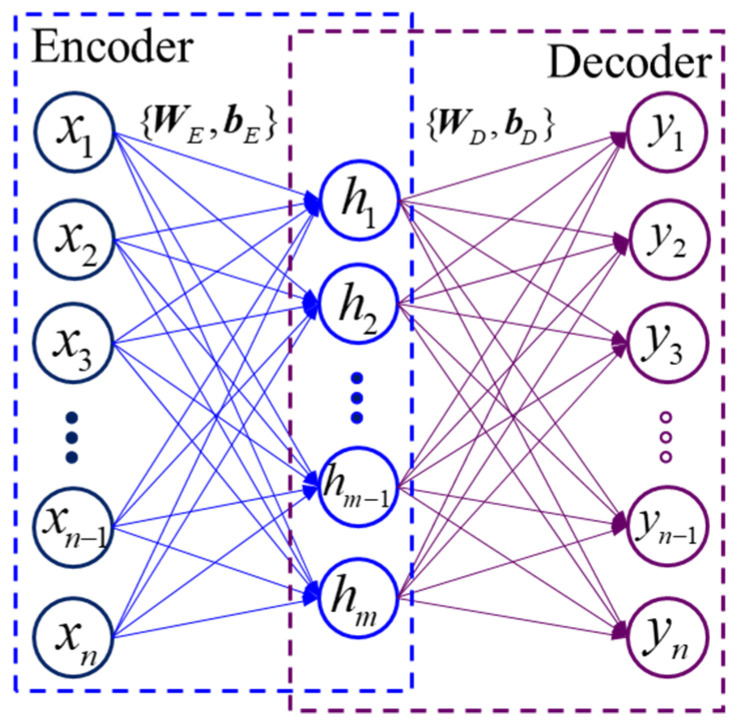
The principle of auto-encoder.

**Figure 2 sensors-22-04943-f002:**
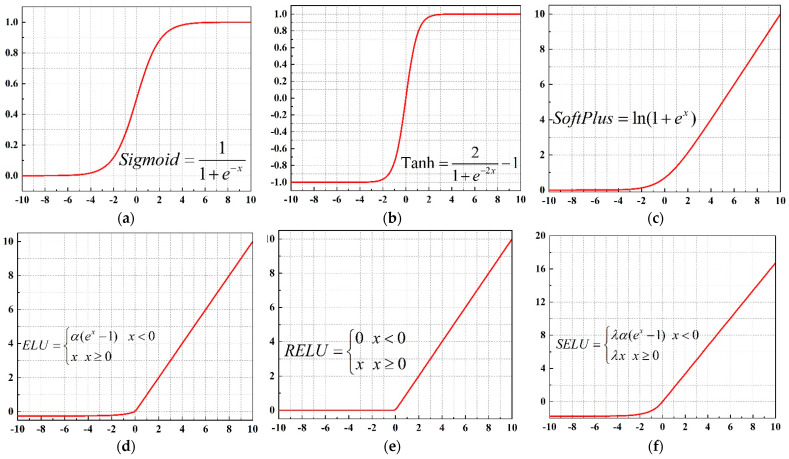
The waveforms of different activation functions. (**a**) Sigmoid; (**b**) Tanh; (**c**) SoftPlus; (**d**) ELU; (**e**) RELU; (**f**) SELU.

**Figure 3 sensors-22-04943-f003:**
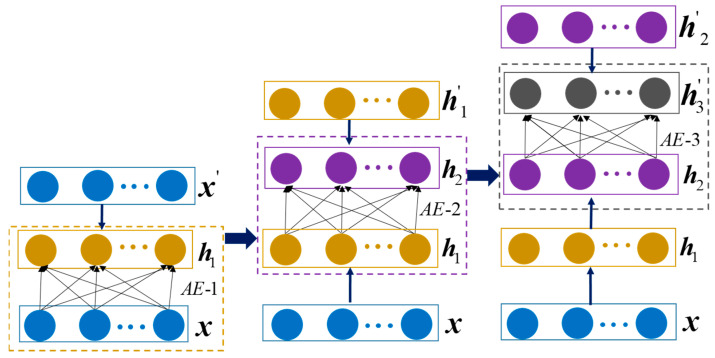
Three-layer stacked auto-encoder.

**Figure 4 sensors-22-04943-f004:**
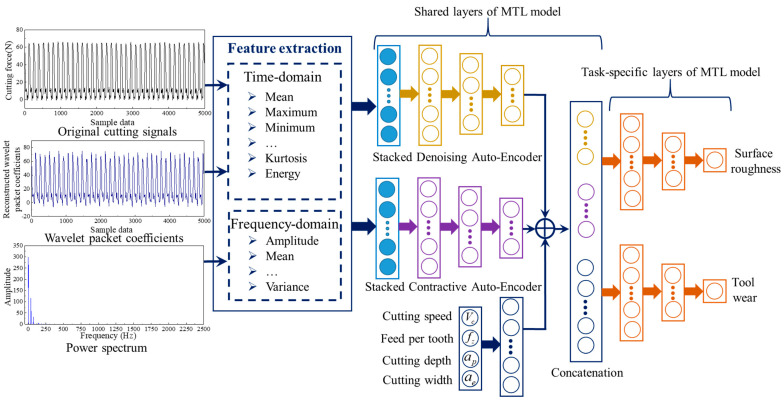
The overall framework of the *MTL* model.

**Figure 5 sensors-22-04943-f005:**
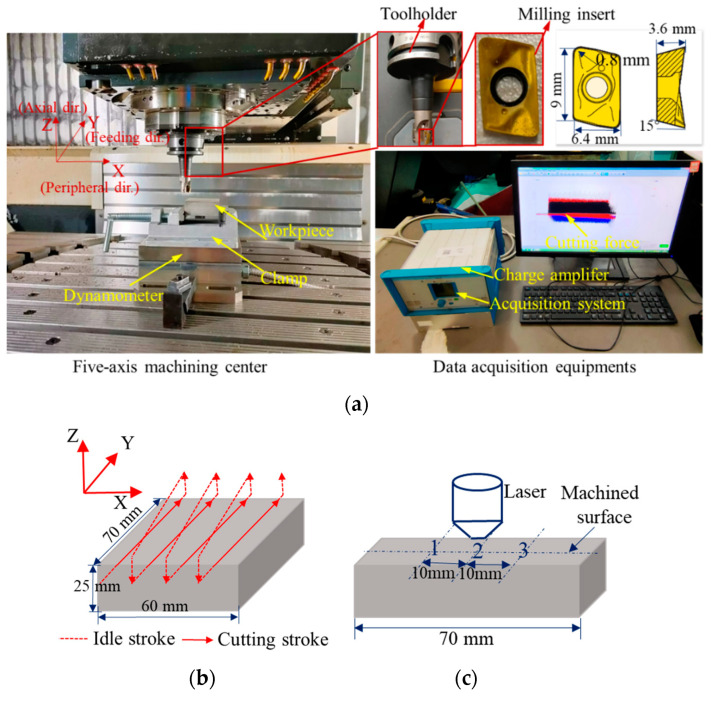
Experimental setup and data acquisition. (**a**) The actual diagram of experimental setup; (**b**) The tool path of cutting; (**c**) The measurement position of surface topography.

**Figure 6 sensors-22-04943-f006:**
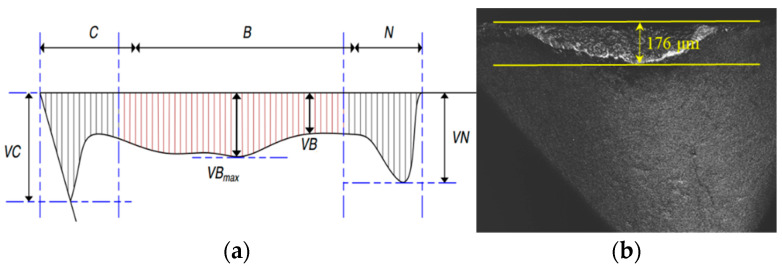
The typical pattern and measurement of flank wear. (**a**) The typical pattern of the flank wear; (**b**) The schematic diagram of flank wear measurement.

**Figure 7 sensors-22-04943-f007:**
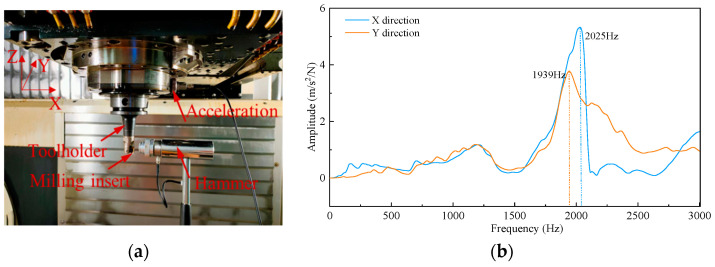
Modal test and frequency response function. (**a**) Modal test; (**b**) Frequency response function of machining system.

**Figure 8 sensors-22-04943-f008:**
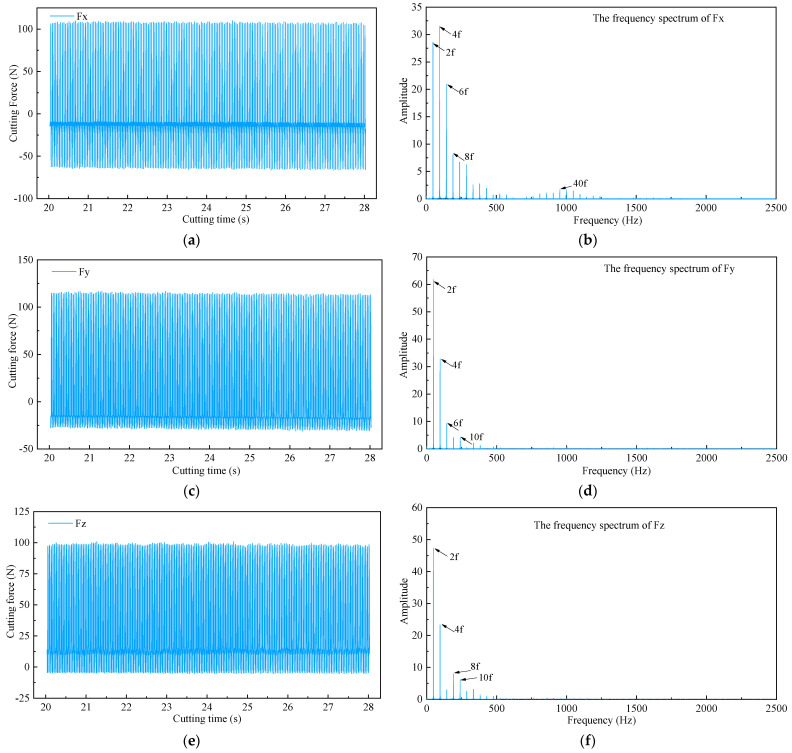
Original cutting force signals and corresponding frequency spectrum. (**a**) Fx component force in X direction; (**b**) The corresponding frequency spectrum of Fx; (**c**) Fy component force in Y direction; (**d**) The corresponding frequency spectrum of Fy; (**e**) Fz component force in Z direction; (**f**) The corresponding frequency spectrum of Fz.

**Figure 9 sensors-22-04943-f009:**
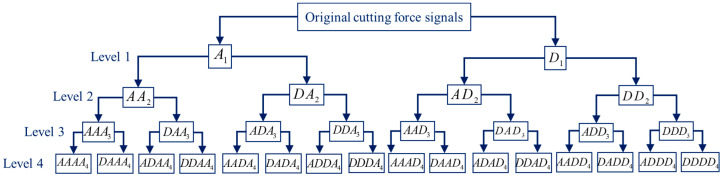
Structure diagram of WPT.

**Figure 10 sensors-22-04943-f010:**
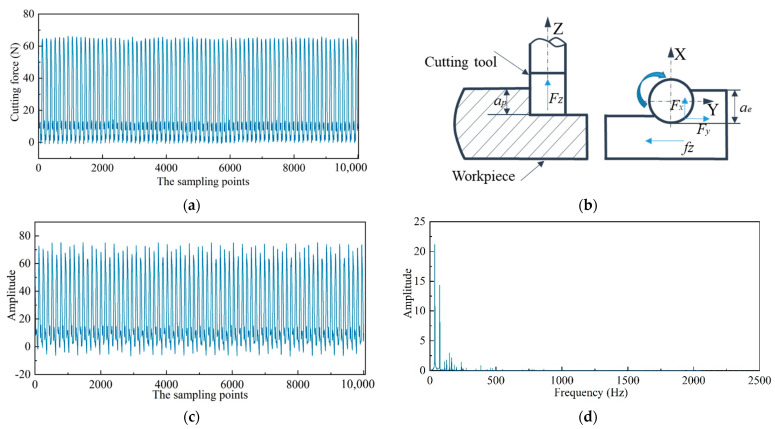
The original cutting force and corresponding wavelet packet transform results of the first four nodes. (**a**) The original cutting force signals; (**b**) The schematic diagram of cutting force; (**c**) The reconstructed wavelet packet coefficients of the first node (**d**) The corresponding power spectrum of the first node; (**e**) The reconstructed wavelet packet coefficients of the second node; (**f**) The corresponding power spectrum of the second node; (**g**) The reconstructed wavelet packet coefficients of the third node; (**h**) The corresponding power spectrum of the third node; (**i**) The reconstructed wavelet packet coefficients of the fourth node; (**j**) The corresponding power spectrum of the fourth node.

**Figure 11 sensors-22-04943-f011:**
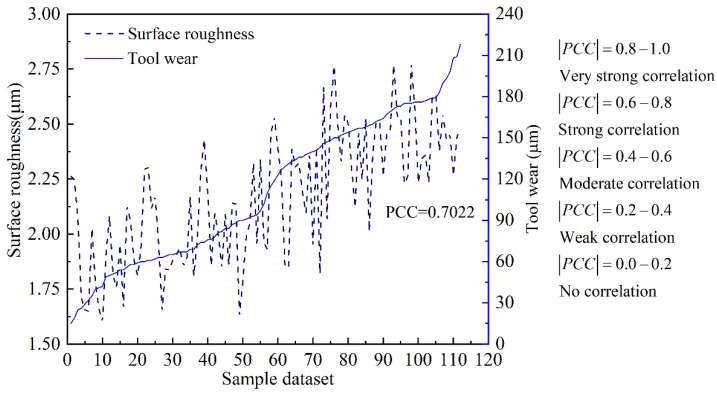
The correlation analysis of surface roughness and tool wear.

**Figure 12 sensors-22-04943-f012:**
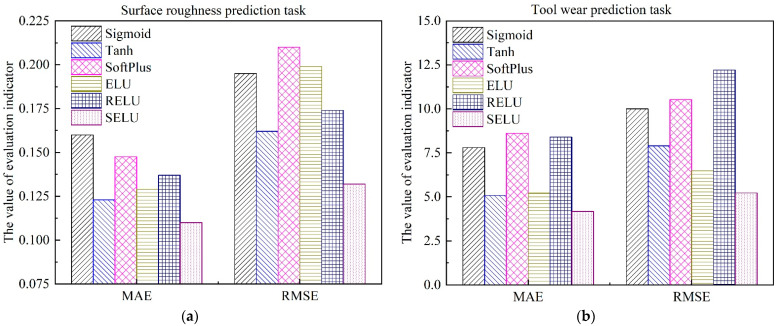
Performance comparison of different activation functions of SDAE, and the activation function of SCAE is RELU. (**a**) Surface roughness; (**b**) Tool wear.

**Figure 13 sensors-22-04943-f013:**
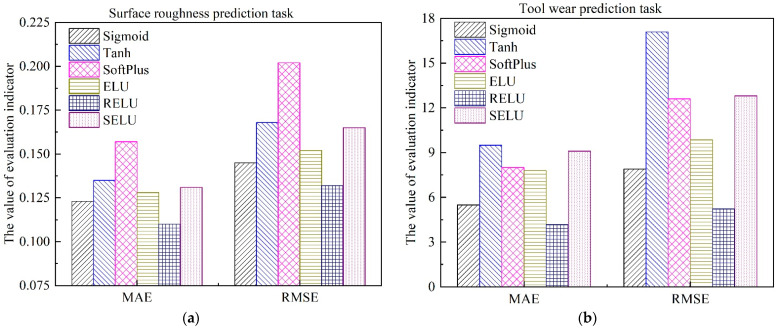
Performance comparison of different activation functions of SCAE, and the activation function of SDAE is SELU. (**a**) Surface roughness; (**b**) Tool wear.

**Figure 14 sensors-22-04943-f014:**
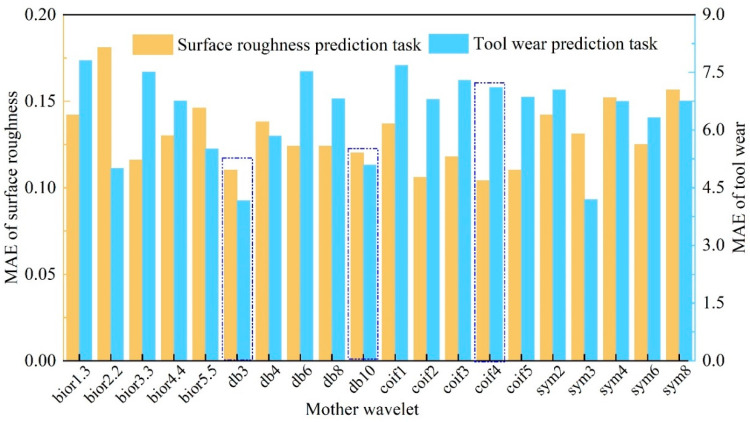
The performance of different mother wavelets.

**Figure 15 sensors-22-04943-f015:**
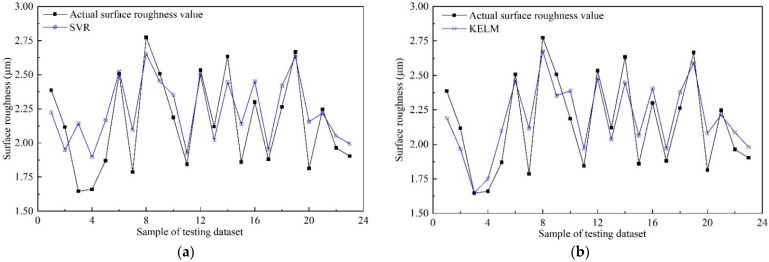
Surface roughness prediction results of different models. (**a**) SVR; (**b**) KELM; (**c**) MTL_SDAE; (**d**) MTL_SCAE; (**e**) STL_PSAE; (**f**) The proposed *MTL* model.

**Figure 16 sensors-22-04943-f016:**
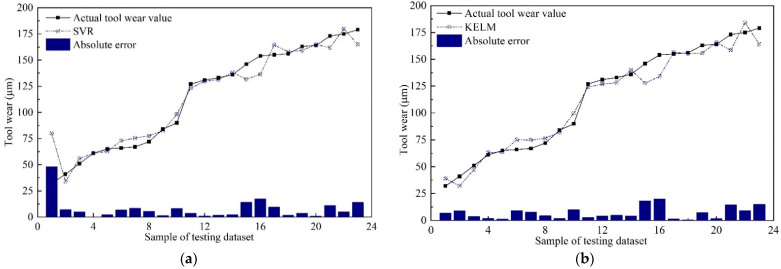
Tool wear prediction results of different models. (**a**) SVR; (**b**) KELM; (**c**) MTL_SDAE; (**d**) MTL_SCAE; (**e**) STL_PSAE; (**f**) The proposed *MTL* model.

**Figure 17 sensors-22-04943-f017:**
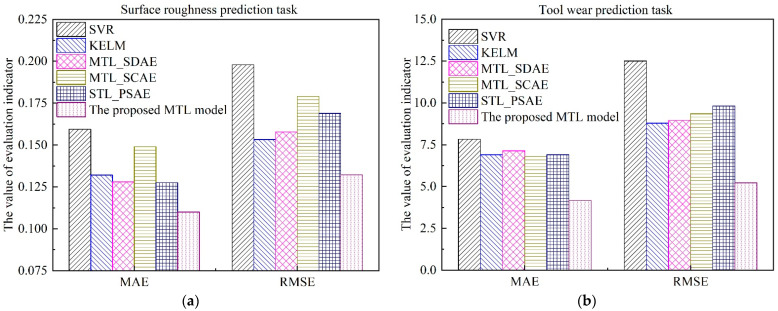
Performance comparison of different models. (**a**) Surface roughness; (**b**) Tool wear.

**Table 1 sensors-22-04943-t001:** Hyper-parameters of the proposed *MTL* model.

Parameters	Value Range	Determined Value
Optimization algorithm	(Adam, SGD, RMSprop)	Adam
Batch size	(8, 16, 32)	16
The number of epochs	(300, 400, 500, 600, 700, 800)	500
Activation function of SDAE	(Sigmoid, Tanh, SoftPlus, ELU, RELU, SELU)	SELU
Activation function of SCAE	(Sigmoid, Tanh, SoftPlus, ELU, RELU, SELU)	RELU
Hidden layer nodes of SDAE	Hidden layer 1: (60, 50, 40), Hidden layer 2: (40, 30, 20), Hidden layer 3: (20, 10, 5)	(50, 30, 10)
Hidden layer nodes of SCAE	Hidden layer 1: (60, 50, 40), Hidden layer 2: (40, 30, 20), Hidden layer 3: (20, 10, 5)	(50, 30, 10)
Nodes of the dense layers for surface roughness prediction	Dense layer 1: (20, 10), Dense layer 2: (10, 5)	(20, 10)
Nodes of the dense layers for tool wear prediction	Dense layer 1: (20, 10), Dense layer 2: (10, 5)	(10, 5)
Nodes of the dense layer for cutting parameters	(10, 5)	5

**Table 2 sensors-22-04943-t002:** Mechanical properties of Haynes 230.

Elasticity Modulus (MPa)	Yield Strength (MPa)	Tensile Strength (MPa)	Poisson Ratio	Hardness (HV)
180	440	842	0.3	175

**Table 3 sensors-22-04943-t003:** Chemical compositions of Haynes 230 (wt%).

Elements	Ni	Cr	W	Mo	Mn	Si	Al
Content	57	20–24	13–15	1.0–3.0	0.3–1.0	0.25–0.75	0.2–0.5

**Table 4 sensors-22-04943-t004:** The geometry of the milling insert.

Number of Cutting Edges	Rake Angle	Clearance Angle	Corner Radius	Insert Thickness	Coating
2	10.5°	15°	0.8 mm	3.6 mm	PVD

**Table 5 sensors-22-04943-t005:** Variation ranges of cutting parameters and tool wear during milling.

Cutting Parameters	Abbreviation	Range	Value Interval	Units
Cutting speed	*Vc*	50–90	10	m/min
Feed per tooth	*f_z_*	0.05–0.1	0.01	mm/tooth
Cutting depth	*a_p_*	0.2–0.4	0.05	mm
Tool wear	*VB* _max_	15–220	/	µm

**Table 6 sensors-22-04943-t006:** The extracted features and corresponding expressions.

Domain	Extracted Features	Expression
Time domain	Mean	$M=\frac{1}{N}\sum_{i=1}^{N}x_{i}$
	Maximum (*Max*) and Minimum (*Min*)	$Max=\max(\left|x_{i}\right|)\&\min(\left|x_{i}\right|)$
	Peak-to-Peak (*PP*)	$PP=\max(\left|x_{i}\right|)-\min(\left|x_{i}\right|)$
	Variance (*Var*)	$Var=\frac{\sum_{i=1}^{N}{\left(x_{i}-M\right)}^{2}}{N-1}$
	Skewness (*Skew*)	$Skew(X)=\frac{\sum_{i=1}^{N}{\left(x_{i}-M\right)}^{3}}{(N-1)s^{3}}$
	Kurtosis (*Kurt*)	$Kurt=\frac{\sum_{i=1}^{N}{\left(x_{i}-M\right)}^{4}}{(N-1)s^{4}}$
	Energy (*E*)	$E=\frac{1}{N}\sum_{i=1}^{N}x_{i}^{2}$
Frequency domain	Amplitude of power spectrum (*Am*)	$Am=\max(P_{i})$
	Mean of power spectrum (*Me*)	$Me=mean(P_{i})$
	Variance of power spectrum (*VPS*)	$VPS=\frac{1}{n-1}\sum_{i=1}^{n}{\left(P_{i}-Me\right)}^{2}$
	Modified equivalent bandwidth (*MEB*)	$MEB=\sqrt{\left(\sum_{i=1}^{n}{\left(f_{i}-\bar{f}\right)}^{2}P_{i}\right)/\sum_{i=1}^{n}P_{i}}$
	Frequency Band Energy (FBE)	$\mathrm{FBE}=\sum_{i=1}^{n}P_{i}{}^{2}$
	Mean Square Frequency (*MSF*)	$MSF=\sum_{i=1}^{n}f_{i}{}^{2}P_{i}/\sum_{i=1}^{n}P_{i}$

Note: *x_i_* represents the original cutting force data or reconstructed wavelet packet coefficients. s is the standard deviation. *f_i_* is the frequency signals obtained by Fourier transform (FT); *P_i_* is the power spectrum of *f_i_*.

**Table 7 sensors-22-04943-t007:** Mother wavelet used in this research.

Type	Family	Order
Biorthogonal	Biorthogonal	bior1.3, bior2.2, bior3.3, bior4.4, bior5.5
Orthogonal	Daubechies	db3, db4, db6, db8, db10
	Coiflet	coif1, coif2, coif3, coif4, coif5
	Symlet	sym2, sym3, sym4, sym6, sym8

**Table 8 sensors-22-04943-t008:** Performance analysis of different models.

	Surface Roughness		Tool Wear	
	*MAE*	*RMSE*	*MAE*	*RMSE*
SVR	0.159	0.198	7.83	12.5
KELM	0.132	0.153	6.90	8.79
MTL_SDAE	0.128	0.158	7.13	8.95
MTL_SCAE	0.149	0.179	6.80	9.35
STL_PSAE	0.128	0.169	6.90	9.81
The proposed *MTL* model	**0.110**	**0.132**	**4.17**	**5.22**

Note: Fonts in **black** indicate the best prediction results.

## Data Availability

Not applicable.

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
