# Peer review of "A Novel Multi-Task Learning Model with PSAE Network for Simultaneous Estimation of Surface Quality and Tool Wear in Milling of Nickel-Based Superalloy Haynes 230"

_sensors, 2022, doi:10.3390/s22134943_

Round 1

Reviewer 1 Report

1.     After each terminology or symbol firstly appear in this paper, it necessary to explain, for example LSSVM, CNN,….

2. The authors declared that “In addition, some other researchers believed that tool wear states would be reflected on the surface topography, so …”. The authors seem to have omitted some references.

3. Please the authors recheck the format of author citation in the manuscript.

4.The content of the paper would have been more significant with meaningful discussion. Authors should highlight upon the meaningful discussion in results and discussion part except presenting the data and observation made thereon. For example, In addition, compared with the most commonly used activation functions Sigmoid and RELU, when SDAE adopted the nonlinear function SELU as the activation function, the estimation accuracy of surface roughness was improved by 31.25% and 19.71%, respectively. And the estimation accuracy of tool wear was improved by 52.87% and 47.44%, respectively, as shown in Figure 9. Why?

Reviewer 2 Report

Authors presented a significant work for estimation of surface quality and tool wear with deep learning and force signals.After reading the submitted manuscript,there are several suggestions for the authors which i am sure will improve the quality of manuscript:

1. I recommend including the techniques/algorithms used in title so that by reading the title, readers will have an idea about the work.For ex: A novel multi-task learning model based on PSAE and SDAE for...............

2. It will be better to include the actual diagram of experimental setup in revised manuscript.

3. Authors mention to utilize WPT for feature extraction but not included which mother wavelet was chosen as there are more than 300 wavelets available. Kindly include separate section and discuss in detail in revised manuscript. Author should refer following journals :

a. https://journals.sagepub.com/doi/abs/10.1177/1077546314520830.

b. https://link.springer.com/article/10.1007/s00170-016-9664-3.

4. What is the motivation to apply parallel stacked auto-encoder based on stacked denoising auto-encoder for roughness prediction. Justifications and description needed based on literature survey.

5. In table 1 hyperparameter of proposed model was mentioned. How the parameters values were decided.

6. Statement in pg.10,line 304-305 "only the time-domain features were extracted from reconstructed wavelet packet coefficients with a low frequency”.Statement is unclear.Further In table 3 extracted features are from wavlet coefficients and frequency coefficients.Is it correct ?

7. How the parameters like kernel function,penalty parameters 'C' for SVR as well as parameters for KELM were estimated. Kindly refer relevant literature.Might be suitable parameter will give better accuracy with  SVR and KELM.

8. Few formatting mistakes were observed.Kindly rectify it.

Reviewer 3 Report

It is generally acknowledged that reliable process condition monitoring based on a single signal feature is not feasible. Therefore, the calculation of a sufficient number of related to the tool and/or process conditions is a key issue in machining monitoring systems. This is obtained through signal processing methods. Azouzi and Guillot apply cutting parameters and two cutting force components for online estimation of surface finish and dimensional deviations. Huang and Chen employ a statistical approach to correlate surface roughness and cutting force in endmilling operations. Abouelatta and Madl develop a method of surface roughness prediction in turning based on cutting parameters and FFT analysis of tool vibrations. Salgado et al. use singular spectrum analysis to decompose the vibration signals for in-process prediction of surface roughness in turning. Song et al. investigate time series analysis of vibration acceleration signals measured during cutting operations for real-time prediction of surface roughness.

The authors proposed an interesting topic that influences the heavy production process.

Below are my critical comments:

1. Why was such a set of cutting parameters adopted (Tab.2)? What was the cutting width?

2. Why was the Sa parameter used to evaluate the surface roughness?

3. Page 11. Why do the authors use the Ra parameter? See page 8 for a description of the Sa parameter.

4. What do the authors understand by the wear factor VB? Please make a drawing. Maybe it is worth using the KE indicator? How was the VB index measured?

5. How can you compare the results (force signals) for different depths of cut for the VB indicator? This is incorrect.

6. How many cutting blades does the milling cutter used in the research have?

7. Figure 7. The authors did not write what force it was. I propose to draw a diagram of the distribution of forces.

8. What was the natural frequency of the machine tool-tool system? This is important when analyzing signals in the frequency domain.

9. Have the extracted features of the signals been normalized?

These are my basic comments. Another problem in reading the article is the lack of description of the markings adopted by the authors. For example, figure 6.

The article requires thorough corrections. It is written incomprehensibly.

Round 2

Reviewer 1 Report

No

Author Response

Thank you again for your review

Reviewer 3 Report

The authors made the suggested changes. The changes are satisfactory. I accept the article as it stands. Thank you.

Author Response

Thank you again for your review